# Wind Catchers: An Element of Passive Ventilation in Hot, Arid and Humid Regions, a Comparative Analysis of Their Design and Function

**Afaq Hyder Chohan * and Jihad Awad**

Department of Architecture, College of Architecture, Art and Design, Ajman University, Ajman P.O. Box 346, United Arab Emirates
* Correspondence: a.chohan@ajman.ac.ae

**Abstract:** This review study circumscribes wind catchers as vernacular zero-energy systems of passive ventilation. The research reviews various types of wind catchers and analyses their design, effectiveness and utility in building design. Furthermore, the study documented some of the technological transformations of wind catchers and their adaptation (functional and symbolic) in various regions. In this context, the complex design data of various wind catchers were appraised, and adaptable design data is compiled in "Geographical and Regional Influences on Wind Catcher Design " and "Performance Evaluation of Wind Catchers" of study. The design analysis uncovered interesting facts about the effectiveness of various types of wind catchers; for example, a wind catcher with one side could be employed only as a wind scoop, whereas a multi-sided wind catcher can work simultaneously as a wind scoop and a heat sink (exhaust). The study also revealed that, in the near past, wind catchers were extensively used for ventilation. However, in modern times they are being adapted as an element of urban and architectural (identity) rather than a functional element. Finally, the end results of this study present candid suggestions for using wind catchers in modern buildings and chalks out blueprints (design guidelines) to adapt wind catchers. Towards the adaptation of wind catchers in modern buildings, this study has worked out 14 key design modifications in different types of wind catcher. Most of these findings are related to improving wind intake, preventing dust and rain penetration, the size of a wind catcher's opening and shaft.

**Keywords:** wind catchers; zero-energy ventilation; wind catcher design; sustainable ventilation

## 1. Introduction

The design of ventilation systems is critical and important for buildings, particularly in housing designs. The designs of wind catchers changes in response to regional climatic conditions, criteria and consideration of ventilation. In the tropical and sub-tropical areas, designers place more emphasis on controlling the level of humidity. However, in other regions, ventilation and tapping of air currents or breezes are important to consider in building design. Additionally, shade and protection from dust storms may have the highest priority in arid and semi-arid regions. In general, the first and second examples are mostly taken into consideration for building design in coastal belts or within range of wind corridors. However, in hot and arid climates, ventilation and cooling through passive means are important for buildings, particularly in suburban or remote areas having little access to conventional power supplies.

Historically, wind catchers have been widely used as an integrated ventilation element designed to allow passive ventilation in buildings of various climatic conditions. A wind catcher is an elevated chimney-like structure built on the rooftops of buildings, which works well in various climatic conditions to harness the atmospheric air and direct it towards the space under it [1]. However, in certain regions of India (Rajasthan), the terms

"wind tower" and "palace of winds" were used for a private (passively ventilated) windy sitting spaces for the royal family [2].

In many regions of the world, inhabitants are adapted to extreme climates and have adapted their external environment and developed ways to minimize indoor temperatures. People have adapted themselves to harsh outer environments, particularly in long and sunny summers when temperatures are high, exceeding tolerance levels. However, a dwelling also turns warm because of solar gain and mitigation through walls, roofs and oriels. Traditional buildings in such areas are good examples displaying innovative techniques to maximize building comfort. One way to clock solar gain is to build in clusters or isolate buildings through narrow lanes; this prevents buildings from gaining direct solar heat and results in less heat mitigation. This eventually affects the temperature inside buildings, and for ventilation inside buildings, tall structures (wind towers) are added to exchange the wind from inside to outside in evening hours or at dusk [3]. The velocity of the wind and temperature variation are important aspects of passive ventilation, and these factors are important for the circulation of air within and outside the building. These factors also determine whether wind catchers are be employed to receive outer wind or to create a stack effect to emit stale internal air. In addition, they are critical for the design of wind catchers, and indeed help to determine a wind catcher's height and the size of its opening [4]. In addition to the availability of a natural wind current, proficient thermal comfort inside a building is also dependent on the building's color and albedo level, conventional shading and protective building elements, nighttime ventilation, insulation and light-colored roof covering. In addition, the proper design of cross-ventilation is also important for passive ventilation inside buildings. In hot and humid climates, an appropriate design of air intakes and exits plays vital role in reducing indoor air temperatures compared to outdoor temperatures [5,6]. Some studies [7–10] have provided detailed information on the design of effective passive ventilation in buildings. These studies considered that double volume inside buildings and their shapes (atria), oriels, roofing and the materials used are important parameters for ventilation design. Moreover, heating and cooling recovery mechanisms (HRV) systems are important in the design of effective passive ventilation inside buildings. These studies stress that the induction of HRV systems in the design of new buildings will ensure energy savings and a satisfactory indoor environment.

During the hottest time of the year, studies have observed that wind catchers with a combination of damp screens successfully worked well to produce thermal comfort by minimizing the buildings internal temperature [10]. In principle, a wind catcher serves two important functions: first, it makes the incoming wind acceptable; and second, it reduces the temperature and causes thermal comfort [11]. Researchers agreed that wind catchers are practically useful in regions far away from metropolitan facilities. Therefore, wind towers are a smart and sustainable method of ventilation inside buildings [12]. Wind catchers are effective for buildings sited with the possibility of poor ventilation, particularly in situations where it is not possible for fresh wind to enter through windows in all parts of a building or buildings with limited openings. In these conditions, it is not possible for wind to enter and change direction, which is required for passive ventilation. In addition, wind catchers are also feasible in buildings with low height or a narrow exterior opening or façade [13].

In certain regions, wind scoops installed at a low height opposite to the direction of the prevailing wind also work as exit points for interior air. In this condition, wind enters from other openings (windows) of the building and leaves through the stack effect from the opening at the roof (low-height wind scoop). In addition, wind catchers also help to create a buoyancy effect in regions of low-velocity air [14,15]. Buoyancy ventilation or stack system ventilation occurs from differences in air density. Fresh cool and dense air enters into a building through its windows and warm and light air leaves through the wind tower. In fact, heavier air forces lighter stale air to exhaust, which generates airflow. The height of the scoop and its distance from the air intake or opening determine the air pressure and velocity of internal air, and more air pressure is created if the intake and exit



openings are at the maximum distance from and opposite to each other [16].In general, a wind catcher works well in regions with wind corridors. However, in low air-velocity regions, the efficiency of a wind catcher can be improved if a low-speed fan can be induced in the wind catcher [17].

Although wind catchers are a traditional method of ventilation from the pre-electricity period, they can be equally good in dense urban developments. Indeed, the high ventilation quality of wind catchers has proved that wind catchers have more advantages than other building oriels. Wind catchers can be proficient and exceed ventilation requirements in dense and polluted urban contexts. A properly designed and modified wind catcher can provide clean air, which is not possible through normal urban windows [18]. The study of retrofitted wind catchers in multi-level (storeys) urban residential buildings with inadequate natural ventilation has shown that the induction of wind catchers has enhanced passive ventilation in buildings. Results have concluded that the installation of wind catchers was particularly suitable for cooling down building interiors in summer days. The study further added that wind catchers proved to be more functional and efficient compared to normal windows [19].

Many regions have adapted and evolved techniques of passive ventilation and thus have different types of wind catchers. Wind catchers are used in coastal belts, and arid and semi-arid regions to accomplish comfortable conditions inside buildings [20–27]. Wind catchers have been in use for centuries, but some studies consider that Iran is the origin of this technology as more complex and highly efficient wind catchers are being used there. Wind catchers (or wind towers) have been commonly used in other regions such as the United Arab Emirates, India, Pakistan, Egypt and Iraq [28]. Based on their design and function, wind catchers have various types, such as one directional, two directional, four directional, hexagonal and octagonal wind catchers. However, the monumental and prominent examples of traditional wind catchers are present in Dowlat Abad garden, Iran [29], the Amir Chakhmaq Mosque, Iran and Alzubair's historical city [30–32]. In addition, new designs of wind catchers are evolving to best fit contemporary lifestyle and architecture; although classical wind catchers are beneficial, contemporary architecture and lifestyle have brought about new customized designs of modern wind catchers [33–36].

The discussion above has revealed the various features of wind catchers in the context ofenergy saving, sustainability, adaptation, climate response and retrofit design. Table 1 shows the key information extracted from above.

**Table 1.** Passive Features of Wind Catchers. Source: Author.

| No | References | Passive Features |
|----|-----------|------------------|
| 1 | [10] | Hybrid wind catchers with a combination of damp screens can produce a cooling effect. |
| 2 | [11] | Wind towers are a smart and sustainable method of ventilation, in regions with no or limited urban facilities. |
| 3 | [13] | Wind catchers are practical to adapt in low-height bonded building sites with minimum exterior façades. |
| 4 | [14,15] | Wind catchers also help to create a buoyancy effect in regions of low-velocity air. |
| 5 | [17] | Performance of wind catchers in low-velocity air can be improved through the integration of low-pressure PV electric fans. |
| 6 | [18] | Clean filtered air can be obtained from well-designed wind catcher, which is not possible through normal urban windows |
| 7 | [33–36] | New designs of wind catchers are evolved to best fit contemporary lifestyle and architecture. |

## 2. Wind Catchers as Sustainable and Energy-Saving Elements

Building examples given by Mazidi et al. (2006) portray that natural wind passing over cooling bodies (water) works as enhanced passive [29] ventilation; this cohesive design can take maximum advantage of available natural conditions and integrated elements of ventilation. Elements of passive ventilation are integral parts of building design and development. These elements are designed to bring natural current without electro-mechanical systems. In general, 24% of global carbon dioxide ($CO_2$) is released by habitable buildings, which consume 40% of total global power [37]. In addition, ventilation and air conditioning (heating and cooling) consume 50% of the total energy required by buildings [38]. These alarming figures can be reduced through adapting the passive techniques of thermal comfort systems, which could also limit the use of air conditioning in buildings [39]. Conventional cooling technologies are a major source of $CO_2$ emissions and environmental pollution in regions with warm climates. Wind catchers can be used as energy-saving and environmentally friendly passive ventilation devices, which are practically suitable for the power-ridden regions of the developing world. Studies have shown that conventional mechanical ventilation in the housing of tropical and hot regions caused the release of 8.6 billion tons of $CO_2$ in 2004, and 11.1 billion and 14.3 billion tons in 2020 and 2030, respectively. These figures narrate that conventional mechanical ventilation and cooling in housing is the second largest source of carbon footprints after heavy industry [40–43].

It is interesting to know that the use of conventional energy in buildings results in more carbon emissions than the transport sector. This indicates that building energy consumption can be reduced through the adaptation of passive techniques of ventilation. Therefore, energy- and environment-conscious people opt for passive ventilation technologies in their new buildings, and in particular the adaptation of wind catchers or towers has substantially increased [44], as shown in Figure 1.

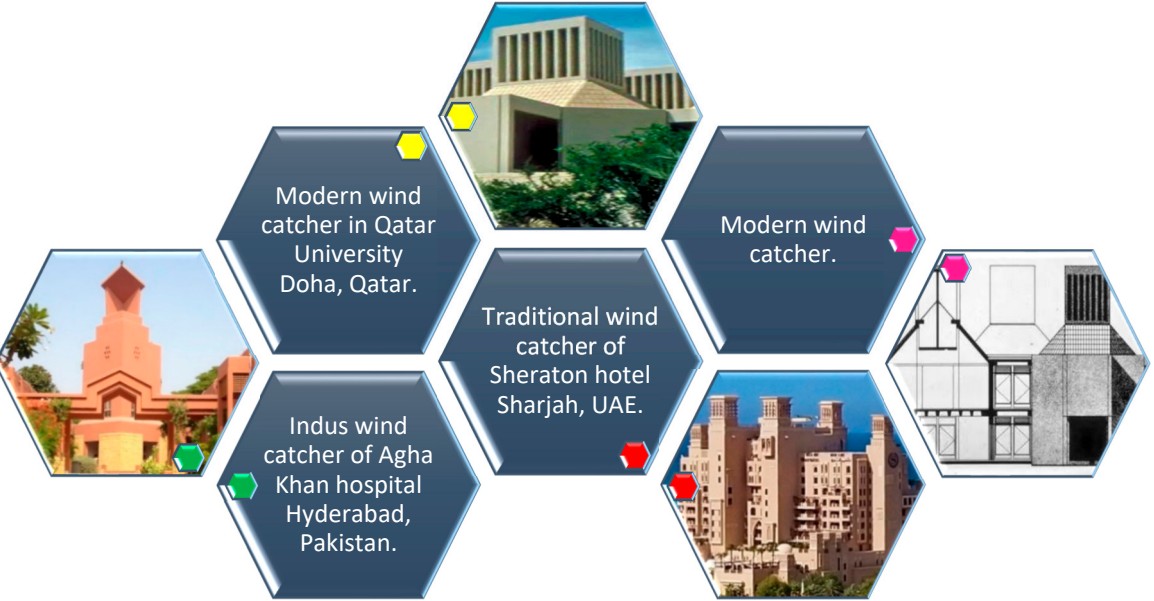

**Figure 1.** Wind catchers in modern architecture. Source: Photos are by authors unless otherwise mentioned (Modern wind catcher in Qatar University Doha, Qatar. Source: [45]; Modern wind catcher. Source: [45]).

The price instability of fossil fuels has mostly resulted in an increase in conventional energy costs, particularly in non-oil-producing countries. Therefore, designers and engineers progressed to introduce alternative renewable energy usage in buildings and adapt natural ventilation techniques [46]. However, passive ventilation in buildings depends on climatic conditions and the design of airflow in buildings. Passive ventilation is considered as a power saving and environmentally friendly technique highly required in the ongoing crisis

of global warming. Wind towers, cross ventilation, stack ventilation, and solar chimneys, are some passive ventilation techniques which can be adapted in buildings. It is proven that these techniques have substantial potential to ensure thermal comfort inside buildings and reduce the demand of air conditioning [47]. In modern building design, wind catchers are drastically ignored as a device of ventilation, whereas in Britain a similar concept to wind catchers has been in use in the last thirty years. In fact, wind catchers are a proven device for combatting the rising carbon emission and global warming issues the world is facing [48–50]. However, in recent years, peoples' awareness about use of natural and renewable energy has significantly increased. Therefore, users insist on building designs that assimilate passive ventilation systems and low power-consuming electro-mechanical devices [51].

The discussion above can be concluded as: Wind catchers are a passive and built-in design element of ventilation, responsible for ventilating buildings with prevailing atmospheric wind. The velocity of wind and temperature variation (inside and outside) are the critical factors for determining the effectiveness of a wind catcher. The design and number of wind catchers depend upon the dimension and number of rooms requiring ventilation. Architectural identity, increasing energy costs and global warming are important factors that encourage the designers to rethink using energy-friendly ventilation techniques as elements in buildings. In this regard, wind catchers and towers have emerged as the most adorable and effective element of passive ventilation. In fact, for a long time, various architectural forms of wind catchers and towers have been in use as an element of non-mechanical indoor ventilation. The architectural characteristics of wind catchers have also abetted designers to use them as objects of aesthetics and identity in new buildings.

## 3. Classification and Design of Classical Wind Catchers

Wind catchers or wind towers are defined as passive elements, projecting out (high) from the top surface of buildings. The most commonly used wind catcher is a tubular structure based on rectangular and square plans. Depending upon regional climatic conditions, wind catchers can be designed with a single opening and single barrel (wind scoop) or a multi-directional multi-barrel wind catcher with heat-basin element. Because of the diverse climatic conditions and multi-directional wind in Iran, remarkable examples of wind catchers have been built and are in use to-date. In some Arab countries such as the UAE, Egypt and KSA. people have adapted two-sided wind catchers (Malqaf and Barajeel) in their buildings. Whereas, in the Indus region, particularly in the cities of Pakistan (Hyderabad and Thatta), one-sided single-barrel wind catchers have been used in residential buildings. However, most studies have been conducted on four-sided wind catchers and louver design. It is worth mentioning that the function and effectiveness of wind catchers are highly influenced by the physical parameters of the wind catchers, such as height, number of openings and the design of the louver [52–55]. Based on their design, wind catchers can be classified into four groups: internal division, cross-sections, number of stores and openings [56]. In addition, further classification of wind catchers can also be made according to number of openings, such as single-side, two-side, four-sides, six-sides and eight-sides [57]. However, on the basis of cross-section (footprint), wind catchers can be classified into five groups, such as square-plan, rectangular-plan, circular-plan, hexagonal-plan, octagonal-plan. However, square and rectangular plans are most commonly used plans in the design of wind catchers [57]. Figure 2 shows the classification of wind catchers.

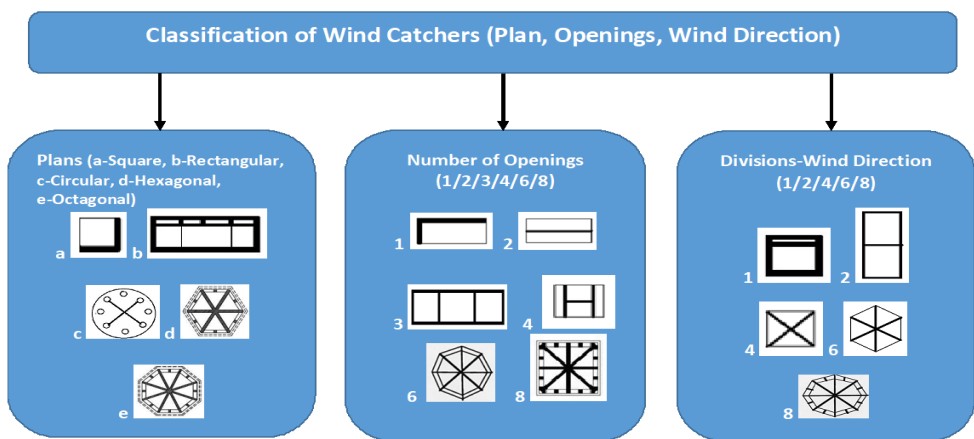

**Figure 2.** Classification of wind catchers. Source: Author.

It is interesting to note that, unlike rectangular and square plans, hexahedral and tetrahedral wind catchers are not very elevated. These types are used for the ventilation of water reservoirs [58,59]. The most critical factors for efficient wind catcher design are the number of openings and height. Variation in wind catcher openings also influences the aesthetic appearance of a wind catcher [60]. In fact, the design of each type of wind catcher is customized according to regional climatic conditions and cannot be adapted in dissimilar regions [61] For example, one-sided wind catchers can work as a wind scoop only and for ventilation they require other forms of openings in the building, whereas, in two-sided/multi-barreled wind catchers, one side will work as a wind intake and the other side will suck stale air from the interior of the building and act as a chimney. Nevertheless, it is important to note that wind catchers with one and two openings bring a higher stream of air inside buildings as compared to wind catchers with four or more openings [62]. Results of various studies have portrayed that the performance and effectiveness of single- and two-sided wind catchers were highest among other competing wind catchers. Therefore, one-sided and two-sided wind catchers are more popular in regions with single-directional wind flow and are considered as economical and sustainable forms of ventilation. The high flow of uninterrupted wind in a one-sided wind catcher is the major reason for its popularity [63–65].

In addition to the number of openings and height of wind catchers, wind velocity is also important for the design of wind catchers. A difference in wind pressure on the surface of a wind catcher plays important role in maintaining the required air supply. Experiments have shown that the wind angle increases with the decline in the transitional angle [53,64,66–72]. Studies of Afshin et al. [66] and Montazeri and Azizian [52] have made experiments with 55° and 50–60° wind catchers and their results show that a minimum ventilation rate occurs in the transition angle, whereas two-sided/multi-barreled wind catchers act reversely in higher angles [49]. The induced airflow rate decreases with the increase in the number of openings. This interesting finding was revealed after testing numerous models of multiple-sided wind catchers in a wind tunnel. Experiments have also shown that the effectiveness of wind catchers is also contingent to a difference in pressure between the inlet and outlet openings [48]. Wind catcher systems can effectively drop the indoor temperature during hot summer if the external (outdoor) air temperature is not high. The results of studies indicate that wind catchers can work efficiently if the temperature difference between inside and outside is between 10–15 °C−15 °C. Studies have concluded that wind catchers are practically useful if the temperature of the heat source (inside house) is 30–32 °C and the external temperature is 20–22 °C, i.e., a difference of 10 °C is desirable for the effective use of wind catchers [73]. Regarding the placement and location of wind catchers, Nejat et al. (2016) added that the ideal placement for a single-sided/single-barreled wind catcher is at the corner of room [74], whereas multidirectional

wind towers would work efficiently if placed in the center of room. In order to summarize the arguments, Table 2 has been produced below.

**Table 2.** Schematic drawings of wind catchers. Source: [44,75].

| Schematic Drawings | Design Criteria | Schematic Drawings | Design Criteria |
|---|---|---|---|
|  | Elevation and plan of 'X' blade four sided wind catcher at square plan. Commonly used in Iran & GCC countries. |  | Elevation and plan of 'X' blade wind catcher at hexagonal plan. Commonly used in Iran, particularly Doulat Abad and Yazd. |
|  | Elevation and plan of 'K' blade multiple sides wind catcher at square plan. Commonly used in Iran & GCC countries. |  | Elevation (view) and plan of single side wind catcher at square plan. Indus wind catcher were used in residential buildings of historical cities of Hyderabad Sindh and Thatta. |
|  | Elevation and plan of 'H' blade four sides wind catcher at square plan, commonly used in Iran. |  | Elevation and plan of single side multi barrel wind catcher at rectangular plan, commonly used in Iran, Egypt & Iraq. |
|  | Elevation and plan of 'I' blade four sided wind catcher at square plan, commonly used in Iran. |  | Elevation and plan of single side single chamber wind catcher at rectangular plan, commonly used in Iran, Egypt & Iraq. |
|  | Elevation and plan of multi sides 'X' blade wind catcher, over circular plan and dome supported by cylindrical columns. A decorative wind catcher used in Yazd, Iran and Sharjah, UAE. |  | Elevation and plan of single side wind catcher, square plan over dome, used in fortress of Herat, Afghanistan. |

The discussion above can be summarised as: The design of wind catchers is customized according to regional climatic conditions and the prevailing wind direction and velocity. It is not practical to adapt a wind catcher for one climatic zone to another. The variety of climatic regions has produced various types of wind catchers, and these can be classified according to their sides and plan (cross section). Square- and rectangular-base, single-barreled/one-sided and double-barreled/two-sided wind catchers are considered suitable for the ventilation of building interiors, whereas circular- and octagonal-base wind catchers are suitable for buildings that require less flow of air inside. Wind catchers work well if there is a variation between the inside and outside temperatures, i.e., the outside air temperature should be lower than the inside temperature of the building. The wind velocity and the height of the tower are the most important factors for the design of a wind catcher.

However, the wind velocity can be increased through adding more height in the shaft (tunnel or barrel) of the wind catcher. Among the various types of wind catchers, the performance of single- and two-sided wind catchers is better than that of four or more directional wind catchers. In fact, the division in the cross section (plan) and a small size of wind intake causes interruption in wind flow and a minimum amount of air enters through a small intake. Figure 3 shows the design and performance factors of various types of wind catchers.

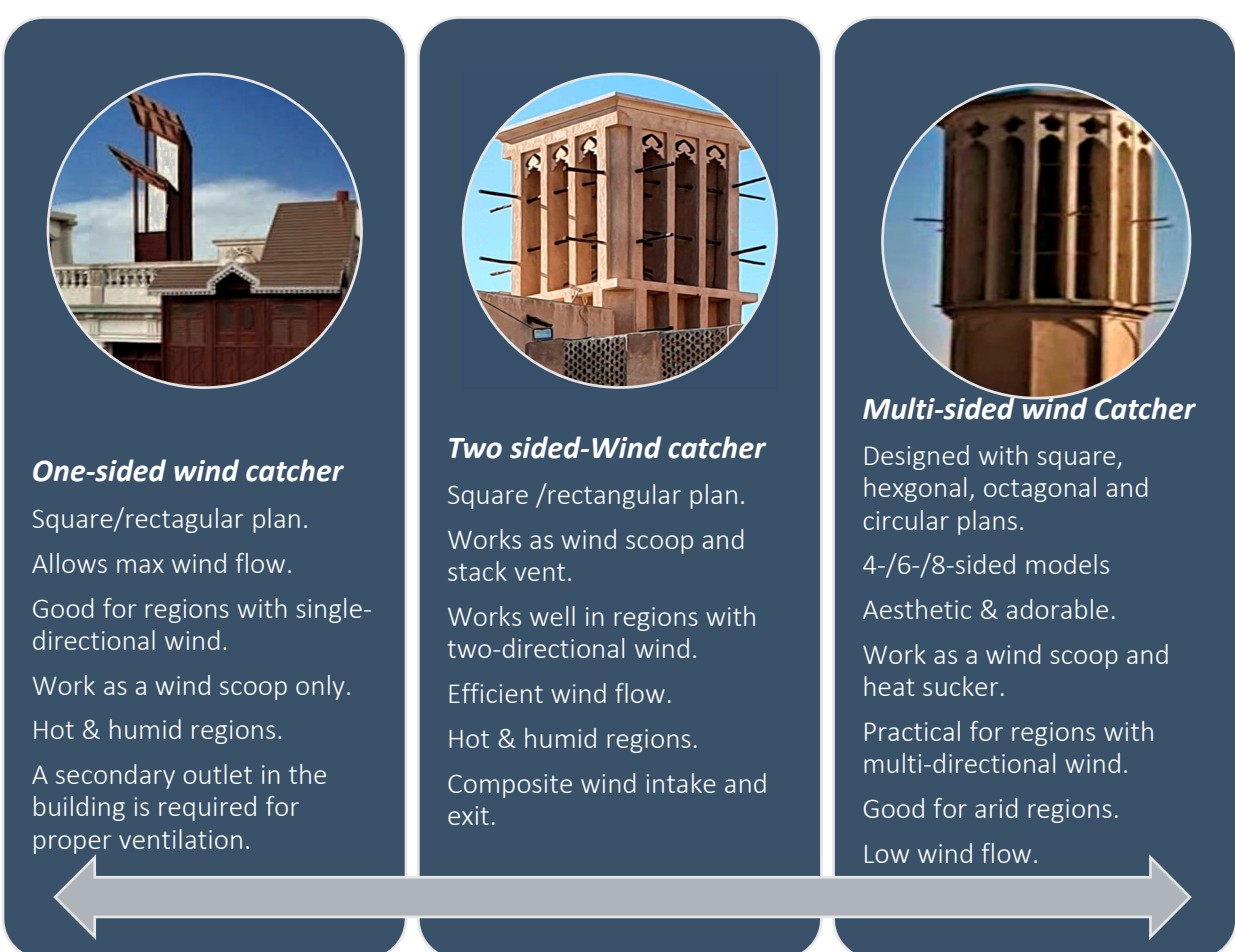

**One-sided wind catcher**

Square/rectagular plan.

Allows max wind flow.

Good for regions with single-directional wind.

Work as a wind scoop only.

Hot & humid regions.

A secondary outlet in the building is required for proper ventilation.

**Two sided-Wind catcher**

Square /rectangular plan.

Works as wind scoop and stack vent.

Works well in regions with two-directional wind.

Efficient wind flow.

Hot & humid regions.

Composite wind intake and exit.

**Multi-sided wind Catcher**

Designed with square, hexgonal, octagonal and circular plans.

4-/6-/8-sided models

Aesthetic & adorable.

Work as a wind scoop and heat sucker.

Practical for regions with multi-directional wind.

Good for arid regions.

Low wind flow.

**Figure 3.** Elements of wind catcher design and performance Source [76].

## 4. Geographical and Regional Influences on Wind Catcher Design

In general, all wind catchers are located at the highest part of a building and require openings facing towards the wind direction. In addition, regional climate, architectural style, construction techniques and materials are important for customizing the function and design of a wind catcher in order to fit in with local conditions. However, the availability of wind currents in summer is the most important factor that determines whether a wind catcher is required in a building or not. Various hot and arid, hot and humid and arid regions in the world have adapted wind catchers for passive ventilation and have their own unique styles and local names for wind catchers.

Iran is considered as the origin of wind catchers and had various adorable models of wind catchers which are best suited for its diverse climate using the term Badgir for wind catchers. In Egypt, wind catchers are called Malqaf, and they have the distinctive characteristic of additive cooling techniques. In the United Arab Emirates (UAE) the word Barjeel is used for wind catchers, and they are unique because of their tower shape and use of projected wooden joists in their design. Because of their geographic location, the Arab and Persian terms for wind catchers (Badgir or Malqaf) are used in Iraq.

Indus wind catchers in Pakistan are called Mangh, and the basic form of wind catcher works as a wind scoop only and has a distinctive projected pointed roof. In Afghanistan, wind catchers are called Badnivil. They are distinctive because of their semicircular roofs. In India, the term Manghu is used in the northwestern part (desert regions) of the country, and the basic design resembles wind catchers in Pakistan.

### 4.1. Wind Catchers of Egypt

An Egyptian wind catcher (Malqaf) is designed with a rectangular plan in a ratio of 1:2, with a triangular structure of 1.5–2-m-high mounting on the roof. Malqafs have been used for rooms and halls and are sometimes coupled with evaporative cooling by providing wet screens in the shaft or a water body under their openings. A traditional form of Malqaf is a two-sided wind catcher, which works as wind scoop and heat sink. However, they are practical for single-directional wind from the northwest direction. The roof of a Malqaf consists of an inclined plate placed at 30°, either supported by side walls or placed on the roof surface, as shown in Figure 4. Malqafs have been widely used in the traditional houses of Cairo and serve up to three-floor buildings [77,78].

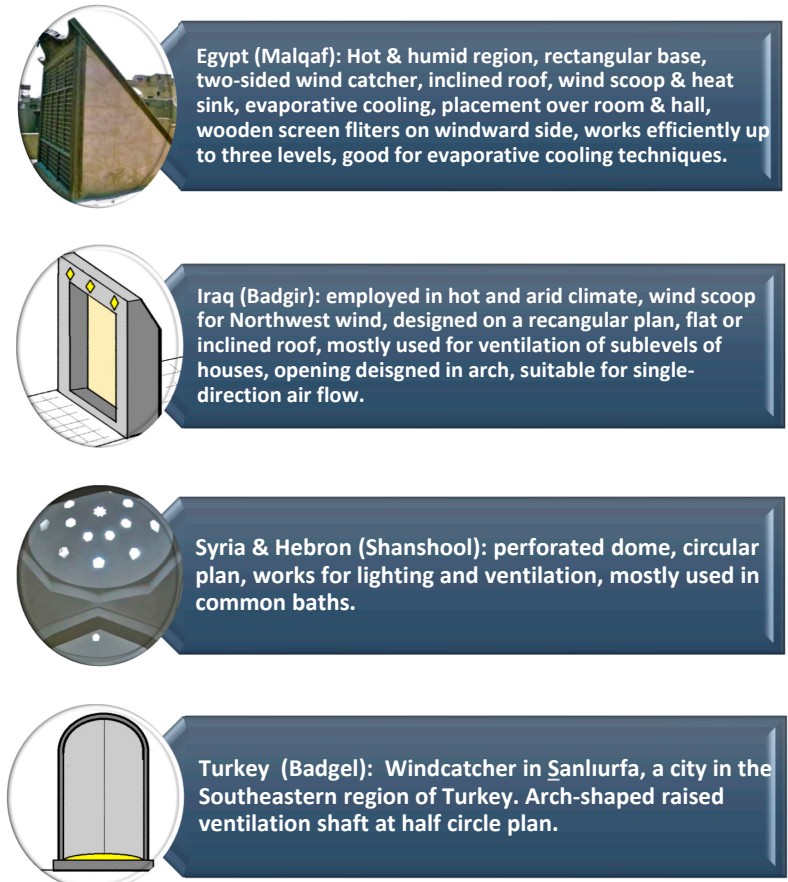

**Figure 4.** Wind catchers (Egypt, Source: [75], Iraq, Syria and Turkey, Source: [79]).

### 4.2. Wind Catchers of Iraq

Wind catchers in Iraq are unique in their appearance, with openings decorated with motifs and a crown-shape lining, available in various forms and shapes. They appear in the form of slots/niches/cavities in a parapet wall, placed either along the wall length next to each other or behind one another. The backs of Iraqi wind catchers (Badgir) are acutely angled to prompt air flow into the rooms. Their openings are designed as arches of various shapes and designs, as shown in Figure 4. Most commonly, pointed, circular and corbelled shapes are used in the wind catchers of traditional houses. Instead of the ground and first floors, wind catchers in Iraq are employed to ventilate sub-floors and basements [77,78,80].

### 4.3. Wind Catchers of Syria

Unlike in Iraq and Egypt, the use of wind catchers in Syria is different. Wind catchers in Syria, termed as Shanshool, are perforated domes placed on the roofs of public baths. According to (Badar [81]), perforations in the domes serve the functions of both lighting and ventilation inside a building (public bath), as shown in Figure 4. Some perforations are positioned in the windward direction to allow fresh air inside and others are placed opposite to the wind direction to act as heat sink, whereas some openings of the perforations are fixed with colored glass to transmit colored light to create ambience in the interior.

### 4.4. Wind Catchers of Turkey

Wind catchers in Turkey (Badgel) are used in the old residential buildings of Sanliurfa city, as shown in Figure 4. The study by (Bekleyen and Melikoğlu [82]) described that a Badgel is an arch-shaped wind catcher based on a 'D' shape circular plan. The shaft of the wind catcher is placed directly on the roof surface up to a height of 1.5 m. The diameter of the circular intake opening ranges between 0.3–0.05 m.

### 4.5. Indus Wind Catchers in Pakistan

Wind catchers in Pakistan (Mangh) were commonly used in the residential buildings of two cities, Thatta and Hyderabad. Both cities are located in a wind corridor region and in the summer, the sea-air current from the southwest direction reaches a velocity up to 26 km/h. The use of Mangh in both cities is almost extinct following the availability of electricity in urban and suburban areas [83]. A Mangh has a square plan in a ratio of 1:1 with two side-walls or plates rising above the roof surface. The roof of a Mangh is a 2–4-m-high element consisting of a 45° inclined plate fixed over two side walls, with its opening oriented to the southwest direction, as shown in Figure 5. The roof of a traditional Mangh is made of timber and plaster or a galvanized metal sheet supported on a timber frame or a brick or mud wall. A Mangh works as wind scoop without the provision of a heat sink, deployed to ventilate the rooms and living areas of a house [84,85].

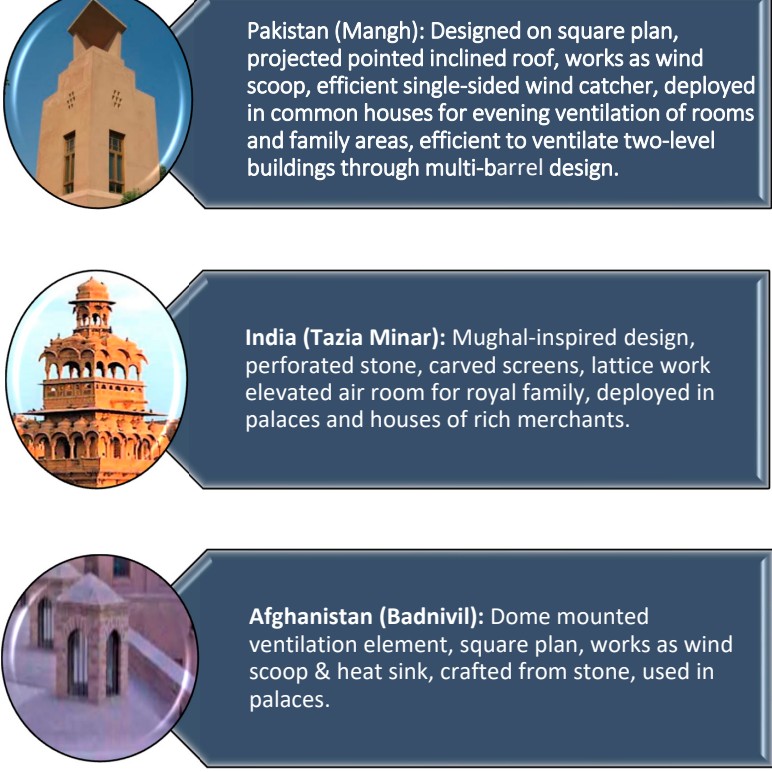

**Figure 5.** Wind catchers (Pakistan, India and Afghanistan, Source: [86]).

### 4.6. Wind Catchers of India

Despite vast architectural design creativity in India, only a few examples could be quoted as being family members of wind catchers. Most recent wind catchers are installed as wind towers in the hostel of University of Jodhpur. Wind catchers are designed to ventilate the common lobby of the hostel, and the shaft of the wind catcher is provided with a wet screen at the windward direction to filter dusty wind and to drop the air temperature. The wind tower proved successful in the summer for maintaining thermal comfort within the active range of the wind tower [87]. However, the earliest form of wind catcher could be traced in the form of wind houses/palaces in Jaipur and Jaisalmer in the northwest of India. Wind houses/palaces are magnificent enclosures, which constitute as the integral space of royal palaces and havelis (big/merchant houses). These enclosures were designed as a family place, particularly for women. The purpose of these structures was to allow royal women to relax in the evening hours in airy spaces enclosed by fine perforated walls and projected windows, as shown in Figure 5. The finest examples of such air rooms are Hawa Mahal, Jaipur and Badal Vilas (Tazia Tower), Jaisalmer, and most of these marvels were designed and crafted by the 'Silawats' (stone carvers) of Rajasthan.

### 4.7. Wind Catchers in Afghanistan

In general, Afghanistan experiences extreme cold in the winter and a moderate summer. Therefore, wind catchers are rarely used in building design. However, Herat fortress has an interesting example of a wind catcher (Badnivil), and its design is simple and works as a wind scoop and heat sink. It is based on a square plan, mounted on a circular masonry dome, and the roof of the wind catcher consists of a low-rise hip roof, as shown in Figure 5. Wind catchers (Badnivil) are deployed in every room with a maximum height of (150 cm), practical for single-directional wind from the northern direction [88].

### 4.8. Wind Catchers in Countries of the Gulf Cooperation Council (GCC)

In the coastal cities of the GCC, wind catchers, or Barajeel, are present in the old houses and living quarters of the local people. Because of the geographic vicinity and trade links with neighboring the country Iran, the design of Barajeels used in the GCC cities such as Dubai and Sharjah (UAE), Manama (Bahrain) and Kuwait City, are quite similar to the wind catchers in Iran Badgeer. Barajeels are based on a square plan in a ratio of 1:1 and are designed to induce multi-directional wind, capable of working simultaneously as a wind scoop and a heat sink, as shown in Figure 6. Traditional wind catchers were made of mud and woven mats made of date leaves and composite tied spines. Later, the design improved through the use of husk-reinforced mud blocks, with detailing and ornamentation on the openings. The roof cover of Barajeels constitute a flat surface made of palm tree mats and were later replaced by a plate made of reinforced mud supported by projected date tree spines [78]. The traditional houses of Bastakia, Dubai, the Kuwait old city district and Bushere, Oman present an antiquity of wind catcher adaptation in the GCC regions [23,89].

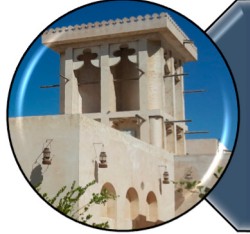

UAE (Barajeel): Good for hot & humid climates, square plan with flat roof, fresco & ornamentaion at opening, ideal for multidirection wind, deployed in rooms and family areas.

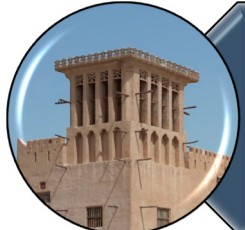

Bahrain (Barajeel): Based on square plan, tower shaft with a variety of opening styles, surface treatment with motifs and texture, good for multi-direction wind.

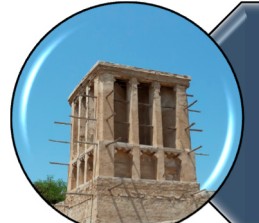

Kuwait (Barajeel): Orginality in material use, flat roof over four-sided masonry posts, mud rendering, multi-direction wind intake.

**Figure 6.** Wind catchers in GCC region. Source: Author.

### 4.9. Wind Catchers of Iran

Iran is the single-most country that possesses a variety in the design and application of wind catchers. In some contexts, it seems that passive ventilation technology was exported from Iran to other regions with similar climatic conditions. Wind catchers in Iran are called Badgir, and some historians believe that wind catchers in Iran were in use since the 4th B.C. Regional designers and craftsmen have remarkably adapted wind catchers according to their climatic condition and aesthetic sense. However, the simplest form of Badgir was traced to 'Tappeh Chakmaq' in northeastern Iran [90]. Wind catchers in Iran were widely used for ventilation. Writings from the 13th century describe that a wide variety of wind catchers were present in the southern, central and northeastern cities of Iran, regions constituting of the cities of Isfahan, Shiraz, Sirjan, Semnan, Teheran, Khorasan and Yazd [23].

A typical Badgir comprises a ventilation shaft (tower) with its lower end opening in a space or room under it, and the other elevated end rising from the roof to catch wind. Unmatched in other regions, Badgirs in Iran are designed on a variety of plans ranging from square, rectangle, hexagonal, octagonal and circular plans and cross sections, as shown in Figure 7. Depending on the availability of wind, Badgir cross-sections are divided into several parts through internal partitions to create air tunnels for incoming air from numerous directions. Badgirs concurrently work as a wind scoop and a heat sink, for example, if wind blows from one direction, the windward openings will be the inlets and the leeward openings will be the outlet, and vice versa. Unlike other regions, wind catchers in Iran have been used for the ventilation of water storage, mosques, caravan places and public parks. The most remarkable examples of Badgirs in Iran can be seen in Ab Anbar, Yazd, the Amir Chaqmaq complex, Aghazadeh House, Golestan Palace, the Ganj Ali Khan Complex and Burujirdi House [17,91].

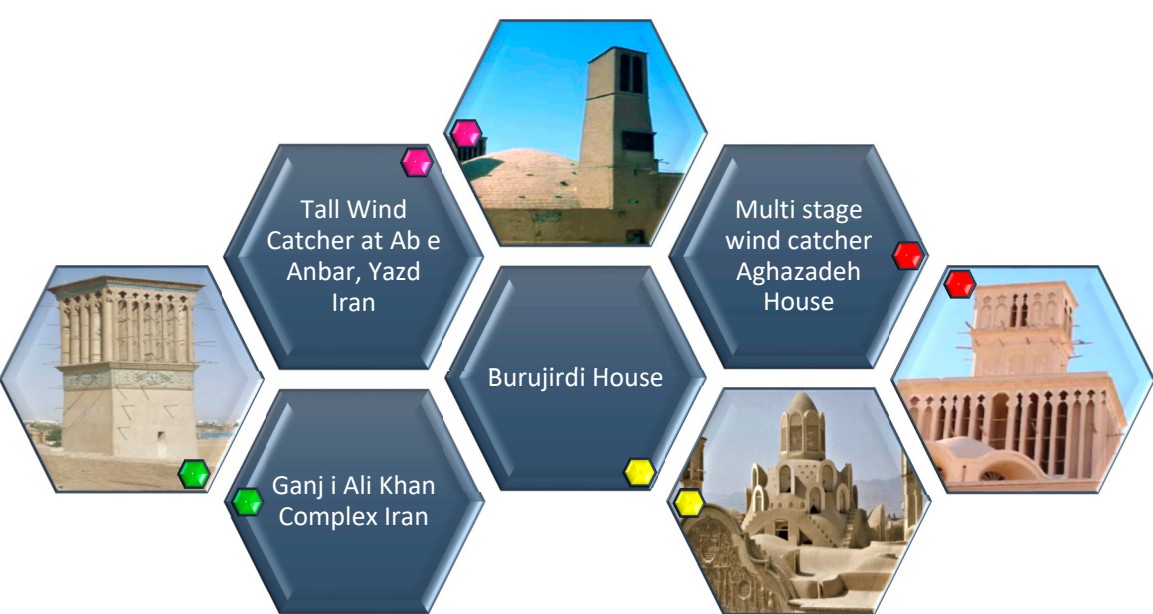

**Figure 7.** Wind catchers in Iran (Ganj i Ali Khan Complex Iran [92], Tall Wind Catcher at Ab e Anbar, Yazd Iran [93], Burujirdi House [94], Multi stage wind catcher Aghazadeh House [95]).

## 5. Performance Evaluation of Wind Catchers

Various studies have been carried out to evaluate the performance of wind catchers through empirical and non-empirical methods. A scaled model is normally used for the empirical mode of research, whereas non-empirical research is carried out through site investigation and case studies [96,97]. Field studies and investigations have proved instrumental observing and measuring performance. However, scaled-model testing is reliable for suggesting and proving the modification of the original design of wind catchers [98,99]. Wind catchers can be compared either through their classification or origin. Based on classification, one-sided wind catchers with an inclined roof provide better air induction compared to one-sided flat roof wind catchers. Studies have further added that using a short up-stream body-facing wind catcher increases its effective intake area and results in higher performance [11]. However, a passive hybrid and energy saving ventilation system is a new buzzword in contemporary times, and various studies have come up with hybrid models (combination of natural ventilation, HRV system and heat pumps). The studies of [100–102] suggest that passive ventilation can be effectively designed with the induction of heat-pump systems in the heating and cooling needs of a building. This type of system can save 51% of conventional energy (electric power) and 49% of renewable energy generation [100–102].

Wind catchers are practically useful not only in maintaining effective ventilation and evaporative cooling, but also in helping to save the bulk of electric power. Studies have shown that wind catchers with evaporative cooling effects can save 52% of conventional energy with improved thermal comfort, with 74–100% satisfaction [103]. Other studies have shown that wind catchers are an ancient method of passive cooling and ventilation, and this passive device works to increase wind velocity and minimize inside temperatures through evaporative cooling [104,105]. Despite sharing a border and culture with Egypt, where wind catchers are used in housing design, the use of wind catchers in Sudan is relatively rare. In certain parts of Sudan, the summer temperature reaches 47–50 °C. In such a dry and arid climate, wind catchers with evaporative cooling can effectively lower inside temperatures by 25–30%.

Passive ventilation through wind catchers minimizes inside temperatures through the increase in wind velocity and its circulation inside a building. In addition to passive cooling, wind catchers also improve the air quality inside a building if combined with the function of a heat basin through two- or multi-sided wind catchers. The performance of

wind catchers is highly dependent on the speed of the outside wind, i.e., the higher the wind velocity, the greater the heat exchange from inside to outside. The other climatic factors that influence the design of wind catchers are relative humidity, water vapor mass fraction and the direction of the blowing wind [13,106–110].

Experimental studies on the number of openings in wind catchers portray that a higher number of openings in wind catchers reduces the intake airflow rate, whereas it is noted that the efficiency of a wind catcher can significantly be improved by means of increasing the difference in pressure between the intake and outlet of a wind catcher [53]. Therefore, the right placement of a one- or two-directional wind catcher is at the corner of room, while the ideal placement for a multi-directional wind tower is at the center of the roof. Studies have further added that the performance of square and rectangular cross-sections (plans) is better than that of other shapes [77]. Kalantar (2009) worked out the role of the material, height, wind speed, atmospheric temperature and humidity in the design of wind catchers. The study added that a 2-m-high wind catcher is effective in reducing the indoor temperature by up to 10–15 °C [111].

Wind catchers in Iran present a complex design and are able to handle diverse climatic conditions. A variety of shapes and planning give them an edge over other competent models of wind catchers. It is significant to note that in traditional housing, wind catchers are placed one after another so that other wind catchers at back can manage any escaping wind from one wind catcher. Wind catchers are still being used as a popular source of ventilation in suburban and countryside settings, particularly during power sharing [112].

In addition, the use of traditional wind catchers in Pakistan is near-extinct in modern buildings, however, certain modified types of wind catchers are designed in residential buildings. The traditional model of wind catchers was a popular mode of ventilation in pre-electricity era houses. This type of wind catcher was instrumental in directing the southwest wind into house without any additive effect of water vapor. Unlike two-faced wind catchers, single-faced Indus wind catchers allow the injection of wind inside a building, and other openings of the building work as an exhaust point [84].

Different regions and their climatic conditions have evolved individual designs of wind catchers to work well in their prevailing climatic and regional conditions. For example, wind catchers used in Pakistan are very different and simple compared to the complex wind catchers used in Iran, the GCC countries and Egypt. Design variation is decisive factor among wind catchers, which influences the performance and effectiveness of wind catchers. Table 3 has been complied to show how climatic factors and regional conditions influence the design of wind catchers.

**Table 3.** Regional climatic conditions and wind catcher design. Source: [75].

| No | Region/Country | Regional Climate | Wind Direction | Cross Section | Height of Vertical Shaft (m) | Size of Plan (m) | Ceiling of Wind Catcher | No of Sides- Wind Intake | Hybrid Function (Evaporative Cooling) |
|---|---|---|---|---|---|---|---|---|---|
| 1 | Iran | Hot & Dry | North East | Square/rectangle/ hexagonal and octagonal. | 3–5 | 0.5 × 0.8, 0.7 × 1.1. | Slope of 45° | Single/Multiple | Yes |
| 2 | Iraq | Hot & Dry | North West | Rectangle | 1.8–2.10 | 0.5 × 0.15 | Slope of 45° | Single side | Yes |
| 3 | GCC countries (UAE, Bahrain, Kuwait, Qatar & Oman) | Hot & Humid | South West/North East | Square/circular | 3–5 | 1 × 1 | Flat/ Slope of 30° | Multiple | Yes |
| 4 | Pakistan | Hot & Humid/Hot & Dry | South West | Square | 2–4 | 1 × 1 | Slope of 45° | Single side | No |
| 5 | Egypt | Hot & Dry | North West | Rectangle | 3–4 | 2 × 0.5 | Slope of 30° | Single side | Yes |
| 6 | Afghanistan | Dry & Mild Hot | North | Square over circular plan | 1–1.2 | 1 × 1 | Slope of 30° | Single side | No |

## 6. Wind Catchers from Functional to Symbolic Elements

Wind catchers are proven and effective elements of passive ventilation, but in modern buildings their usage is minimized. However, in recent years, designers have introduced wind catchers in modern buildings with a modified function and shape. In the Middle East (mainly in the GCC), wind catchers can be seen as part of buildings, regardless of their use

as traditional wind catchers. Universities have used a modified shape of wind catchers over certain spaces at campuses, as shown in Figure 8. Large-scale functional wind catchers have been used for the first time in the Torrent Research Centre, Ahmadabad India, designed by Abhikram Architects. To achieve low energy consumption, the wind catcher system was designed based on Passive Downdraft Evaporative Cooling (PDEC). This system significantly decreased the expulsion of carbon monoxide into the atmosphere and approximately 200 metric tons of air conditioning was saved [113]. In Sharjah (UAE), an imposing wind catcher is part of the Gold Souq (Central Market). An adorable traditional wind catcher used in the hotel and market of Madinat Jumerah in Dubai and a modern modified wind catcher used in the museum of modern arts in Tehran are a few examples that portray the inclusive use of wind catchers in modern architecture and building design, as shown in Figure 8. The wind catchers in these modern buildings portray the visual appearance of a ventilation system, even if the building is centrally air conditioned. Perhaps the concept of these designs is to redefine the element of traditional architecture in contemporary and modern architecture.

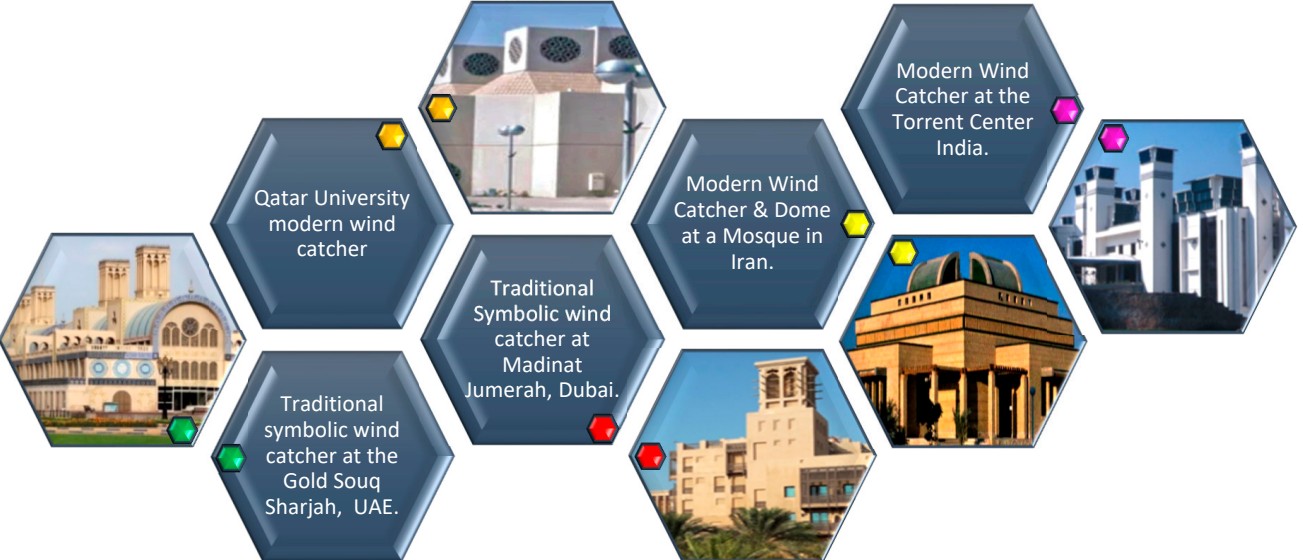

**Figure 8.** Wind catchers in modern architecture (Qatar University modern wind catcher [45], Modern Wind Catcher & Dome at a Mosque in Iran [114], Modern Wind Catcher at the Torrent Center India [115]).

In addition to these, buildings in Egypt are designed with shaftless wind catchers, particularly those designed by Hassan Fathy, a famous native architect. In his designs, a new type of wind catcher was used without a shaft or tower and placed directly over a roof opening. In addition, in some cases, a perforated barrel vault was designed for ventilation [116]. His projects with wind catchers included family residences and neighborhood housing, as shown in Figure 9.

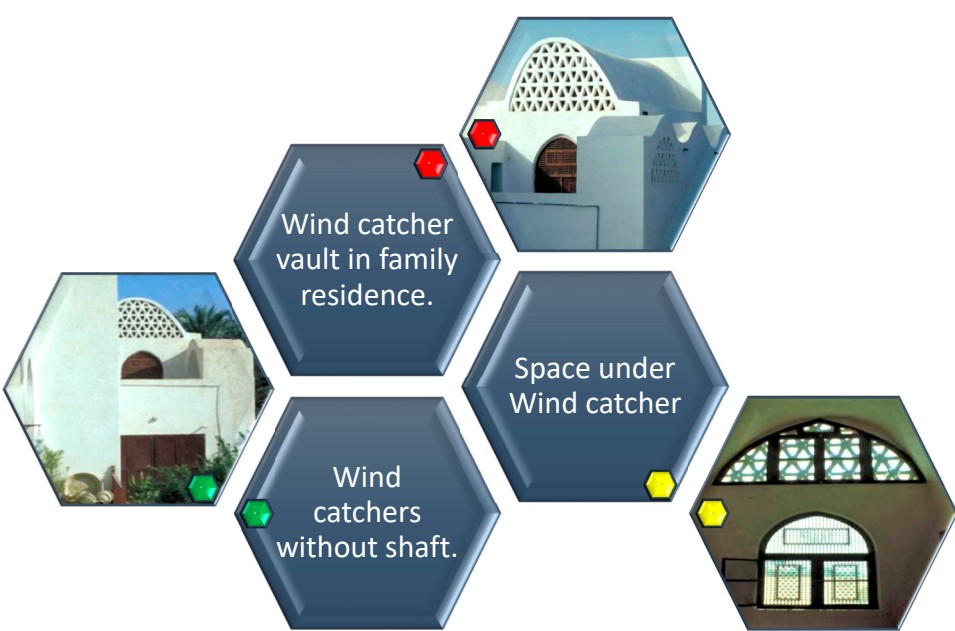

**Figure 9.** Wind catchers in housing architecture. Source: [116].

## 7. Design Features of Contemporary Wind Catchers

There are several reasons that prevent the functional use of wind catchers in present-day buildings. In this context, one of the major reasons is the decrease in the level of thermal adaptation among people. In the pre-electricity era, the thermal resistance level of people was quite high, and passive ventilation through wind catchers was enough to meet their comfort needs. However, the original design of wind catchers cannot withstand the changing lifestyle and requirements of thermal comfort. However, the performance of wind catchers can be improved through scientific experimentation. A study on modified wind catchers have shown that a modified wind catcher with built-in shower cooling was employed to ventilate interior space. The study added that a modified double-barreled wind catcher was effective and performed well in buildings consisting of two floors [117]. However, the function of passive ventilation devices actively depends upon the conditions of the local climate. Modified and hybrid passive ventilation can be developed through experiments and simulations on low air velocity, built-in evaporative cooling devices and the adaption of vapor fans in passive ventilation [118,119]. Studies have suggested that modified wind catchers can be deployed in modern residential, small commercial and institutional buildings for thermal comfort and to conserve energy. Wind catchers are particularly important for saving energy during the peak hours of summer [75,120]. In the context of thermal comfort and energy saving, wind catchers work better than roof gardens and ponds. Wind catchers are recognized as a contributing element for reducing the energy requirements for cooling during the summer [121]. However, passive ventilation alone cannot ensure thermal comfort. Besides selecting an appropriate method for thermal comfort, the design of a building's peripheral external wall (envelope) is potentially important for ensuring comfort inside a building. It is significant to note that proper building orientation and thermal insulation can save 20% of energy use and provide natural ventilation; the choice of facade material can also save an additional 30% of energy use. Studies have further stressed that hybrid systems are also effective and useful in buildings that require low energy and depend on the local climate [118,122–124]. In addition, the studies of [125–127] have elaborated the phenomena of passive ventilation and thermal comfort in context of dry, arid and humid regions, particularly in the United Arab Emirates (UAE). Studies have stressed that traditional and passive design techniques can be used as a benchmark for ensuring thermal comfort in the design of modern house in the UAE.

## 8. Conclusions

In the past, wind catchers were the only available mode of ensuring thermal comfort in the pre-electricity era. Wind catchers remained a popular and affordable element of thermal comfort, and indeed were a metaphor for modern (non-refrigerant) air coolers. However, because of their limitations, wind catchers cannot be utterly equated with modern air coolers. Crammed with the tranquil lifestyle in urban areas, people have misjudged the qualities and advantages of this passive ventilation method. In general, people, both in urban and suburban settings, have lost interest in using wind catchers. In addition, urban congestion and the increase in the wind-obstructing heights of buildings also have discouraged both users and designers from using this ventilation system.

The discussion above revealed that hybrid ventilation is a new concept of ventilation in new building design, i.e., the combination of natural and mechanical heating and cooling techniques. This new integrated design could help to achieve several benchmarks such as maintaining indoor temperatures, grasping heat energy recovery (HRV), preventing the reliance on single systems of ventilation and being energy friendly.

Experiments on various types of wind catchers have proved that modified wind catchers can still be used as hybrid ventilation systems in modern buildings. The literature review in this research, particularly from [4,13,44,60,65,75,84,117,119,128], has revealed that the design of wind catchers can be revitalized through suggested guidelines and modifications, as mentioned in Table 4 below.

**Table 4.** Suggestions for wind catcher design modification. Source: Author.

| Wind Catcher Type | Modification | Issue | Design Suggestion |
|---|---|---|---|
| Single/two sided | Upper opening of shaft | Wind-borne dust | In order to avoid dusty gust into rooms, wind catcher design (single/two) can be modified by adding a dust catcher cavity in the upper opening of the shaft, along the walls of the wind catcher. |
| Square and rectangular | Cross section | Wind velocity | Square and rectangular plans (cross sections) of wind catchers are more efficient that other types of wind catchers. |
| Circular/cylindrical type | Limited height | Wind intake | Circular plan wind catchers are considered as the least efficient wind catchers because of the constraints of the incident angle and limited height. |
| All types | Wind factor | Angle of incoming wind | In general, all types of wind catchers work effectively if the angle of incoming (intake) wind is zero degrees. |
| Rectangular wind catcher | Modified intake opening through louvers | Adjustment of incoming wind angle | A rectangular wind catcher can be modified and upgraded with multiple louvers, which perform very well if incoming wind propels at a zero-degree angle. |
| All types | Height of shaft | Capture wind | Determination of wind speed in the region is important for the effectiveness of wind catcher, and in certain regions an acceptable level of wind velocity can be captured at a height of eight meters. |
| Multi opening wind catchers | No of sides | Least efficient | Provision of the maximum height of a wind catcher is more important than the number of openings in a wind catcher. |
| Single or two side wind catcher | Smart sensors and PV powered hoppers at wind intake opening | Block hot outer wind | Smart temperature sensors and electro-mechanical shutters or hoppers can be fixed at the lower opening of a wind catcher shaft to prevent the intake of outside air with higher temperature. |

**Table 4.** *Cont.*

| Wind Catcher Type | Modification | Issue | Design Suggestion |
|---|---|---|---|
| All types of wind catcher | Size of intake opening | Effective wind intake | The size of a wind catcher opening should be in an acceptable ratio to floors, i.e., the opening of wind catcher should be 8–12% of the floor area |
| All Types | Temp requirement for better performance | General requirement | Wind catchers are highly efficient in regions that have substantial variation in day and night temperature, for example, in day time the mercury reaches 35–40 degree centigrade, and in the evening (sunset) temperature drops to 25–30 degrees centigrade. |
| All types | Temp requirement for better performance | General requirement | Wind catchers can work effectively if the maximum variation between the indoor and outdoor temperature is 10–15 °C. |
| One and two sides | Modification of shaft lower opening | Prevention of rain penetration | In certain regions, monsoon rains in summer disturb the function of wind catchers and causes soaking through the shaft. This can minimized by the collection of soak-away water through making a drip at edges of the lower opening of the shaft or adding a 'U' channel to collect water. |
| One and two sides | PV powered, low-velocity vapor fan system at wind intake opening | Low velocity hot air | In arid regions with low air velocity, a wind catcher can be modified with a PV (photovoltaic) low-speed fan and vapor system (mist) at the lower end of the shaft to enhance wind temperature and draft inside the building. |

This study considers that these design tips will help the intended users and designers to improve the design of traditional wind catchers as per regional climatic conditions. These guidelines will drastically improve the working and efficiency of wind catchers and ensure their easy adaptation in the designs of new buildings. This research considers that wind catchers in the housing sector could be considered as a way forward to save energy resources. In general, because of the growing population worldwide, housing design and construction are always in higher demand. The case studies discussed in this study show that wind catchers work effectively in dwellings, particularly in single-family houses (detached, attached/semi-attached). An institutional obligation to adapt and implement passive ventilation in housing design could lead to a substantial saving in power consumption. Regarding the cost of wind catchers, market analysis has shown that the construction cost of wind catchers is ≤0.3% of the total project cost. A typical 3–4 bed single-family house in the UAE cost 0.8–1.2 million AED. The area required by a typical wind catcher would range from 8–12% of the floor area, i.e., 16–18 sq. ft. At present, in the UAE, construction costs range from 80–120 AED/sq. ft, i.e., the construction cost of a wind catcher (RCC) in the UAE could cost 1280–1920 AED/wind catcher (approx.).

**Author Contributions:** Being the first author A.H.C. and the correspondent author has complied the relevant literature and conducted the comparative analysis. In addition, A.H.C. was responsible for the draft of the manuscript, concluding the research and resolution of the output. Being the second author, J.A. has analyzed the relevance of the literature and provided the technical data. In addition, J.A. conducted the proof-reading of the paper and provided the pictures sourced as Author. All authors have read and agreed to the published version of the manuscript.

**Funding:** This research received no external funding.

**Institutional Review Board Statement:** Not applicable.

**Informed Consent Statement:** Not applicable.

**Data Availability Statement:** Not applicable.

**Acknowledgments:** The authors would like to express their gratitude to Ajman University for APC support and the Healthy and Sustainable Built Environment Research Center at Ajman University for providing an excellent research environment.

**Conflicts of Interest:** The authors declare no conflict of interest.

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
