# Peer review of "Wind Catchers: An Element of Passive Ventilation in Hot, Arid and Humid Regions, a Comparative Analysis of Their Design and Function"

_sustainability, doi:10.3390/su141711088_

Round 1
Reviewer 1 Report
This paper reviews various types of wind catchers and analyses their design, effectiveness, and utility in building design. The article is highly needed to summarize the wind catchers’ typologies in the Moslems world. However, I recommend modifying the following before accepting the paper.
1. - As the study has neither shown any simulation nor experiments done by the authors, the title should reflect the theme included in the study, which is ‘types of wind catchers and analyses their design, effectiveness, and utility in building design”. A zero-Energy Ventilation Element is not justified at all. I recommend removing it from the title.
2. - In the abstract, the main findings of this research need to be highlighted.
3. -In the introduction part, the relationship between passive ventilation and energy saving is widely discussed but needs to be summarized in Table form with respective references.
4. -The text needs some typo corrections.
5. -It would be good to add a few more recent references related to climate and passive design in the UAE. I recommend the following:
The impact of passive design strategies on cooling loads of buildings in temperate climate.
- The Impact of Passive Design Strategies on a Villa in Hot Climate: Case in Sharjah, UAE
-A study of the impact of major Urban Heat Island factors in a hot climate courtyard: The case of the University of Sharjah, UAE
-Application of the analytic hierarchy process to developing sustainability criteria and assessing heritage and modern buildings in the UAE.
- Climatically and culturally responsive typologies using mathematical permutation
6. - Some figures can be replaced by better quality images. Sources of some photos need to be mentioned.
7. - In conclusion, it would be enriching to improve the given Table 1 and precisely discuss the way forward to adapt wind catchers in popular buildings and climates i.e. single-family house.

Author Response
Dear Reviewer 1
Thanks for your suggested comments, I have fulfilled all comments and attached herewith the response report. Your comments were practically helpful to improve the quality of paper. I hope you will be satisfied with my corrections I have made based on your comments.
Regards
Dr. Afaq Hyder

Reviewer 2 Report
Reviewing report
The wind catcher was a passive cooling technology. Introduction of wind catcher through enough literature reading has been presented, not limited to the list of types, local characteristics, performance evaluation, and finally listed the wind catcher and other building energy saving. Various types of wind catchers and their design, effectiveness and applications in building design were reviewed.
Some points were still raised after reviewing this manuscript,
1. The specific schematic diagrams of various types of wind catchers should be presented. Such diagrams could facilitate readers to read and understand.
2. Although the wind catchers in different regions have their own characteristics, they should not only incorporate with the regional culture, but also carry out self-specific analysis with reference to the regional climate and thermal structures.
3. There was a lack of a certain amount of data graphs, and numerical graphs are suggested to enrich present review.
4. The whole paper was not very logically organized. There were too many places in the article that emphasize the energy efficiency of passive ventilation or wind catchers, whether or not it was relevant to the topic of the subsection, such as line 145-160 in part2, line 210-213 in part 3, line 399-416 in part 5.
5. The Introduction section was unstructured and illogical.
6. In part 2, line 145-149 was a statement of the economic reasons for the spread of natural ventilation technology, followed by a statement that much of the spread was due to environmental reasons, in line with the previous paragraphs, and that the economic reasons did not make sense in the middle.
7. In part 2, line 161-164 “Discussion above can be concluded as, wind catchers are placed high above the roof .... depends upon the dimension and number of rooms required to be ventilated.” This section did not conclude with the discussion in this paragraph.
8. The structure of the article was not so clear. Please try to optimize the structure of its subheadings.
9. Section 4 was also a note on design and it could be used as 3.1.
10. Finally, this manuscript should be careful edited by someone with expertise in technical English editing, and paying particular attentions on English grammars, spellings, and sentence structures so that the goals and results of the study are clear to the reader.
Author Response
Dear Reviewer 2
Thanks for your comments, I have complied with all comments , attached please find the response report. Your comments have practically helped to improve the quality of paper. I hope you will be satisfied with corrections and actions I have taken based on your comments.
Regards
Dr. Afaq Hyder

Reviewer 3 Report
The research reviews various types of wind catchers and analyses their design, 8 effectiveness and utility in building design.
The author consulted a large number of literatures and made a detailed analysis.
Some suggestion and comments is :
1) Do you need to include the regional restrictions in the title?
It seems that these areas are mainly:hot & arid, hot & humid, arid
2)Will this technology bring additional design and construction costs to the building?
3) Figure 2 is rough. It is suggested to beautify and add some illustrative pictures
4) There are many discussion in Part3, and many influencing factors are mentioned. However, only Figure 2 and Figure 3 are illustrate.
It is suggested to add table to summarize the total conclusion to guide application.
The same suggestion is made for part 4 and part5 to author.
5) Figre4,5,6 and Figure7, all of them is intorducation of different country. Maybe these figures are suitable for being merged.
6) In the conclusion, part7, the application restrictions and design modification are summed up. Howver, these content should be introduced in another part instead of conclusion. The conclusion should shown all analysis results.
Author Response
Dear Reviewer 3
Thanks for your comments and suggestions. I have complied with all comments, attached please find the response report. Your comments and suggestions have practically helped to improve the quality of paper. I hope you will be satisfied with corrections and actions I have taken.
Kind Regards
Dr. Afaq Hyder

Round 2
Reviewer 2 Report
revision could be accepted for present journal. More references could be cited, and English expressions could be further enhanced.